# Post-hoc Discriminator Guidance For Data-efficient Image Generation Via Annealing Langevin Dynamics

## Abstract

The proposed method, post-hoc discriminator guidance (PDG) aims to take an alternate route for Nash non-equilibrium issue in GANs' training. This method introduces an additional discriminator that gives explicit supervision with regard to gradient of density ratio $\nabla_x \log \frac{p_r(x)}{p_f(x)}$ between real and fake probability density function, steering the sample path towards more realistic regions in a post-hoc way. We train the discriminator after adversarial optimization, making post-hoc discriminator training stable and fast to converge. In generation process, annealing Langevin dynamics sampling with density ratio score reduces the Kullback-Leibler divergence between the true and generated samples. Given an optimal discriminator, the method can improve the sampling quality of various off-the-shelf models on the web without retraining required. Extensive experiments validate the advancements and effectiveness of PDG on content-varying data-limited datasets.

## 1 Introduction

The building of GANs (Goodfellow et al. (2014); Karras et al. (2020b)) is mostly in place, but there are still two dark clouds, one is Nash non-equilibrium, another is discrete data generation. Difficulty in reaching Nash equilibrium during training is an inherent property of GANs for minimax optimization, like the spin of a particle. Besides, the presence of zero real part and eigenvalues with a large imageinary part in associated gradient vector field prevents GANs from convergence (Mescheder et al. (2017)). Furthermore, the disadvantage of adversarial optimization is more obvious in the case of limited data (Karras et al. (2020a)), even causing mode collapse. Hence, many works are proposed recently for this problem by adapting the network architecture (Radford (2015); Liu et al. (2020); Wang et al. (2022)), redesigning objective function and regularization (Tseng et al. (2021); Arjovsky et al. (2017); Gulrajani et al. (2017); Salimans et al. (2016)), and refining training strategies (Jiang et al. (2021)).

In this paper, inspired by classifier guidance in diffusion models (Dhariwal & Nichol (2021)), we focus on a non-game sampling method specially designed for GANs, namely providing the correction gradient information with regard to $C_\theta(x) = \nabla_x \log \frac{p_r(x)}{p_f(x)}$ during generator sampling by a post-hoc well-trained discriminator, while annealing Langevin dynamics ensures that the direction of travel is in the direction of $\frac{p_r(x)}{p_f(x)}$ with high probability. This unilateral optimization is more stable than adversarial optimization, our method just keeps the pre-trained model fixed and steers generation path towards more realistic regions with post-hoc discriminator estimating whether the sample trajectory is reliatic or not.

At the same time, our proposed method is convenient and simple and can be applied to various off-the-shelf models publicly available on the web, after all, just training an additional discriminator always costs less computational resources and costs than re-training the entire GAN network. In this way, the proposed method assists practitioners and researchers who only have limited training samples or computational resources, to access more high-fidelity samples only with a pre-trained generator and a optimally-trained post-hoc discriminator. In experiments, the training of post-hoc discriminator is a stable minimization problem and fast to converge, ablation experiments on various

content-varying datasets verify the effectiveness and advantages of it. Our contributions can be summarized as follows:

- A GAN-specialized post-hoc discriminator guidance method is proposed to add a correction score $C_\theta(x_t)$ on generated images by a way of annealing Langevin dynamics, steering sample trajectory to more realistic regions.

- We prove that the adjusted model score $s_\theta(x_t) + \omega_c C_\theta(x_t)$ is equivalent to the gradient of logarithm of the real data distribution, namely $\nabla \log p_r(x_t)$, when discriminator guidance weight coefficient $\omega_c = 1$.

- Experiments and theoretical demonstration validate the effectiveness and advantages of our proposed method, the discriminator-guided samples are indeed closer to real data than non-guided samples.

## 2 PROBLEM FORMULATION

The mainstream GANs nowadays use stochastic gradient descent (SGD) to optimize the generator and discriminator, similar to the EM algorithm which firstly fixes $p(z|x)$ and optimizes $p(x|z)$. However, the SGD does not find the Nash equilibrium point because the Jacobian matrix of its associated gradient vector field has eigenvalues with zero real part and eigenvalues with large imaginary part.

**Proposition 1** *When the Jacobian of the associated gradient vector field $v'(x)$ has only eigenvalues of the negative real part, let $h \to 0$, SGD can receive local optimal solutions $\overline{x}$.*

In order to prove the above proposition, we start from classical theorem for convergence of fixed-point iteration and throw a theorem.

**Theorem 1** *Assume $F : \mathbb{R}^n \to \mathbb{R}^n$ a continuously differentiable function on open subset $\Omega$, $\overline{x} \in \Omega$, let $F(\overline{x}) = \overline{x}$ and $\left\| det(F'(\overline{x})) \right\| < 1$, exist an open neighborhood $U$ of $\overline{x}$, making $\forall\ x_0 \in U$, $F^{(k)}(x_0) \to \overline{x}$ at least with a linear convergence rate.*

According to Lagrange's mean value theorem, for $\forall x_1, x_2 \in \Omega$, have

$$F(x_1) - F(x_2) = F'(x_1 - x_2) \tag{1}$$

where $F'$ denotes the Jacobian matrix of $F$. For the upper bound of norm $\|F(x_1) - F(x_2)\|$ we throw a lemma.

**Lemma 1** (Bertsekas (1997)) *For the induced norm $\|\cdot\|$ of any matrix $A$, have $\lim_{s \to \infty} \|A^s\|^{\frac{1}{s}} = \rho(A) \le \|a\|$, so given $\forall\ \xi > 0$, $\exists\ \|A\|$, making $\|A\| = \rho(A) + \xi$.*

The above lemma tells us that there exists norm $\|\cdot\|$ and corresponding open sphere $S$ centered at $\overline{x}$, making norm $\left\| F' \right\| < 1 - \xi$ within sphere $S$. So

$$\|F(x_1) - F(x_2)\| \le \left\| F' \right\| \|x_1 - x_2\| \le (1 - \xi) \|x_1 - x_2\| \tag{2}$$

obviously $F$ is a contraction mapping on $[x_1, x_2]$, contraction mapping principle tells us there exists Banach's fixed point, and the convergence rate of the iterative scheme is at least linear.

On the other hand, we need to construct the following function to reach $det(F')$, where $det(\cdot)$ denotes eigenvalue.

$$F(x) = x + hK(x) \tag{3}$$

where $h \in \mathbb{R}$, $h > 0$, besides, finding fixed point $\overline{x}$ is equivalent to solving non-linear equation $K(x) = 0$, hence, the Jacobian matrix of $F$ in Eq. 3 can be formulated as

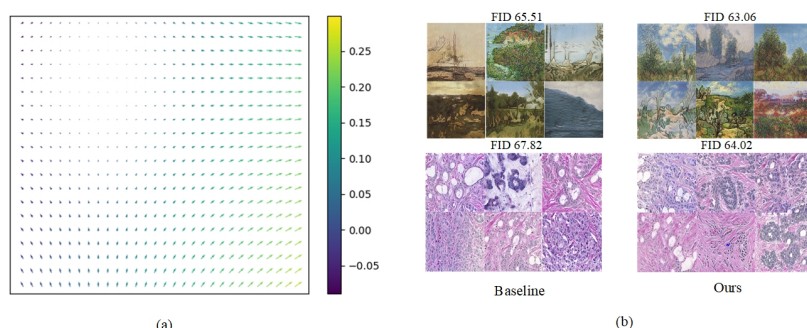

Figure 1: (a) An visualization example of discriminator guidance score $C_\theta(x)$, we plot the magnitude and direction of the gradient at each pixel location. This score offers enhanced mode convergence, by providing distinctive gradient information on whather a sample path is realistic or not via annealing Langevin dynamics, darker colors represent gradients closer to negative values. (b) Samples from ArtPainting (Liu et al. (2020)) and BrecaHAD (Aksac et al. (2019)) datasets, we improve ProtoGAN (Yang et al. (2023)) on former by $2.45$ FID value, MoCA (Li et al. (2022)) on latter by $3.8$ FID value, elaborating that our method promotes the quality of model sampling in a stable min-min way instead of max-min adversarial way.

$$F^{'}(x) = I + hK^{'}(x) \tag{4}$$

However, $det(K^{'})$ is complicated, we throw a new lemma.

**Lemma 2** *Assume $A \in \mathbb{R}^{n \times n}$, only has eigenvalues of the negative real part. If $h > 0$, the eigenvalue of $I + hA$ is within unit circle, the following holds if and only if any eigenvalue $\lambda$ of the matrix $A$.*

$$h < \frac{1}{\aleph(\lambda)} \frac{2}{1 + (\frac{\wp(\lambda)}{\aleph(\lambda)})^2} \tag{5}$$

where $\aleph(\lambda)$ denotes the real part of $\lambda$, $\wp(\lambda)$ imaginary part of $\lambda$.

let $A$'s eigenvalue $\lambda = -a + bi$, where $a > 0$, so $det(I + hA) = det(1 + h\lambda)$. We can deduce that module $|1 + h\lambda| < 1$, namely

$$|1 + h\lambda|^2 = (1 - ha)^2 + h^2b^2 = 1 - 2ah + h^2a^2 + h^2b^b < 1 \tag{6}$$

reshuffle the above equation

$$h < \frac{2a}{a^2 + b^2} = \frac{2a^{-1}}{1 + (\frac{b}{a})^2} \tag{7}$$

According to Eq. 6, it is obvious that the real part of eigenvalue of $A$ is positive number when $-a > 0$, eigenvalue of $I + hA$ is out of unit circle. In this case, $F$ does not satisfy contraction mapping principle, so it doesn't converge.

From Eq. 5, there are two factors impacting step size $h$, one is maximum of $\aleph(\lambda)$, another is maximum of $r = \frac{\wp(\lambda)}{\aleph(\lambda)}$. When $r \to \infty$, $h \to 0$ and $det(F^{'}) \to 1$, making the convergence rate very slow according to **Theorem 1**.

## 3  METHODOLOGY

Except Nash non-equilibrium, GANs are also confronted by mode collapse and discriminator over-fitting. In short, it is difficult for GANs to reach the Nash equilibrium point under the constraint of adversarial loss. This raises a question: how can we close the Kullback-Leibler distance between the true and generated distribution without adversarial loss? The solution is simply a post-hoc discriminator guided approach, namely, a discriminator is trained with generated images and real images to guide the gradient of logarithm of density ratio $\frac{p_r(x)}{p_f(x)}$.

### 3.1  GENERAL FRAMEWORK

GANs try to train a generative model $q(x|z)$ which maps $q(z) \sim \mathcal{N}(z; 0, I)$ to real data distribution $p_r(x)$, where

$$q(x|z) = \delta(x - g(z)), q(x) = \int q(x|z)q(z)dz \tag{8}$$

$\delta(\cdot)$ and $g(z)$ denote Dirac function and generator respectively. Because the trajectory from $z$ to $x$ is deterministic, we ignore the posterior $p(z|x)$. Let us introduce binary hidden variable $y$ into $q(x)$, have

$$q(x, y) = \begin{cases} p_r(x)p_1, y = 1 \\ q(x)p_0, y = 0 \end{cases} \tag{9}$$

here $p_1 + p_0 = 1$, define another joint distribution function $p(x, y) = p(x, y)\tilde{p}(x)$, minimize $D_{KL}(q(x, y)||p(x, y))$

$$\begin{aligned} D_{KL}(q(x, y)||p(x, y)) &= \int p_r(x)p_1 \log \frac{p_r(x)p_1}{p(1|x)p_r(x)} + \int q(x)p_0 \log \frac{q(x)p_0}{p(0|x)p_r(x)} \\ &\sim \int p_r(x)p_1 \log \frac{1}{p(1|x)} + \int q(x) \log \frac{q(x)}{p(0|x)p_r(x)} \end{aligned} \tag{10}$$

when $D_{KL}(q(x, y)||p(x, y)) \to 0$, $q(x, y) \to p(x, y)$ and

$$p_1 p_r(x) + p_0 q(x) = \sum_y q(x, y) \to \sum_y p(x, y) = p_r(x) \tag{11}$$

Define $p(1|x) = d(x, \theta)$, $p(0|x) = 1 - d(x, \theta)$ in Eq. 10, where $d(\cdot, \theta)$ denotes discriminator network. Similar to the EM algorithm, we firstly fix $g(z)$, namely fixing $q(x)$, then optimize $p(y|x)$, the two steps are alternated, simplify Eq. 10

$$D(x, \theta) := argmin[-\mathbb{E}_{x \sim p_r(x)}[\log d(x, \theta)] - \mathbb{E}_{x \sim q(x)}[\log (1 - d(x, \theta))]] \tag{12}$$

### 3.2  PREDICTION AND CORRECTION

Firstly we sample images from a trained generator, then train a discriminator with the generated images and real images using binary cross entropy according to Eq. 12. The correction term gradient on $t$ can be formulated as

$$C_\theta(x_t) = \nabla \log \frac{d(x_t, \theta)}{1 - d(x_t, \theta)} \tag{13}$$

Having acquired this density-ratio score, we can use Langevin-MCMC if $\epsilon \to 0$, $t \to +\infty$.

$$x_t \leftarrow x_{t-1} + \frac{\epsilon^2}{2} C_\theta(x_{t-1}) + \epsilon z^{t-1}, t = 0, 1, \cdots, t - 1 \tag{14}$$

this SGD-like method makes sure $q(g(z)) \rightarrow p_r(x)$ in a kind of Langevin annealing way.

### 3.3 GRADIENT PENALTY

For stable Langevin annealing, it is necessary to do Lipschitz regularization on the partial derivative of $d(x, \theta)$, denoting $\frac{\partial d(x, \theta)}{\partial x}$. Given sample $x_t$ and $x'_t$, the Euclidean distance between them is defined as $\left\| x_t - x'_t \right\|$ which is proportional to

$$\left\| d(x_t, \theta) - d(x'_t, \theta) \right\| \tag{15}$$

when $x'_t \rightarrow x_t$, $\left\| d(x_t, \theta) - d(x'_t, \theta) \right\| \rightarrow 0$, namely satisfying the following equation

$$\left\| d(x_t, \theta) - d(x'_t, \theta) \right\| \leq C \left\| x_t - x'_t \right\|^\alpha \tag{16}$$

where $\alpha > 0$, the simplest equation is when $\alpha = 1$

$$\left\| d(x_t, \theta) - d(x'_t, \theta) \right\| \leq C \left\| x_t - x'_t \right\| \tag{17}$$

only in this way can Euclidean distance between samples satisfy stability requirement. Meanwhile, the sufficient condition that the $d(x_t, \theta)$ follows Lipschitz regularization can be formulate as

$$\left\| \frac{\partial d(x, \theta)}{\partial x} \right\| \leq C \tag{18}$$

where $C$ denotes constant and $C > 0$.

---

**Algorithm 1** Training

---

**Input:** $\Upsilon = \{x_1, \ldots, x_N\}$ from real world, $\wp = \{\hat{x}_1, \ldots, \hat{x}_M\}$ from $G(z)$, Total iteration.
**Output:** $D_\theta$
  1: **while** Current iteration $<$ Total iteration **do**
  2:     Sample $x_1, \ldots, x_{B/2}$ from the real dataset $\Upsilon$
  3:     Sample $x_{B/2+1}, \ldots, x_B$ from the generated dataset $\wp$
  4:     Calculate $\hat{\mathcal{L}}_\theta \leftarrow -\sum_{i=1}^{B/2} \log d(x, \theta) - \sum_{i=B/2+1}^{B} \log (1 - d(x, \theta))$
  5:     Update $\theta \leftarrow \theta - \frac{\partial \hat{\mathcal{L}}_\theta}{\partial \theta}$
  6: **end while**

---

**Algorithm 2** Sampling

---

**Input:** $G$, $z \sim N(0, 1)$, $d(x, \theta)$, Total steps.
  1: $x_g = G(z)$
  2: **for** $t = 1$ to Total steps do **do**
  3:     clip $d(x_g, \theta)$ to a certain range.
  4:     $C_\theta(x_g) \leftarrow -\nabla \log \frac{d(x_g, \theta)}{1 - d(x_g, \theta)}$
  5:     $x_g \leftarrow x_g + \omega_c C_\theta(x_g) + \epsilon z$
  6: **end for**
  7: **return** $x_g$

---

### 3.4 KULLBACK-LEIBLER DIVERGENCE ANALYSIS

In this section, we mainly discuss the question whether $q_{\theta\infty}$ is closer to the data distribution $p(r)$ than $q_\theta$ if we define generated data distribution $p_r$ and after-discriminator-guidance distribution $q_\infty$, from the perspective of statistical divergence in the context of optimization.

**Proposition 2** *Suppose discriminator $D(x, \theta)$ is fully trained, then*

$$D_{KL}(p_r||q_{\theta\infty}) + \Phi_{\theta\infty} \leq D_{KL}(p_r||q_\theta) \tag{19}$$

where $\Phi_{\theta\infty} = \frac{1}{2} \int \mathbb{E}_{p_r} \left[ \|C_{\theta\infty}(x) - C_\varphi(x)\|^2 \right] dx$, we estimate $C_\varphi(x)$ with a neural discriminator $d_\varphi$. In terms of well-trained discriminator $d_\varphi$ , namely minimizing Eq. 12, get

$$C_\varphi(x_t) \approx C_\theta(x_t) = \nabla \log \frac{d(x_t, \theta)}{1 - d(x_t, \theta)} \tag{20}$$

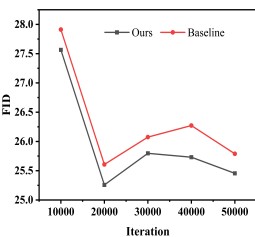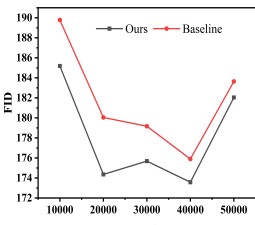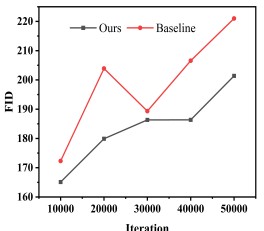

(a) FID (↓) comparison on Grumpy Cat (Zhao et al. (2020)).

(b) FID (↓) comparison on Fauvism (Liu et al. (2020)).

(c) FID (↓) comparison on Shells (Karras et al. (2020a)).

Figure 2: We carry out ablation experiments on generator models with different number of iterations, the results prove that our method is able to improve the sampling quality at all time ends.

In a statistical sense, our model narrows the KL distance between generated and real data as is shown in Fig. 2, verifying the correctness of **Proposition 2** experimentally, although the Kullback-Leibler divergence error contains an empirical risk term $\Phi_{\theta\infty}$ in Eq. 19. On the other hand, the premise of boosting is that the discriminator $D(x, \theta)$ should be trained adequately. When the discriminator is completely blind, namely $d(x_{t-1}, \theta) = 1/2$ in Eq. 20, there is no gradient signals from $C_\theta(x_t)$, this is reflected in the fact that an undertrained discriminator does not change the distribution of images and gets zero FID value gain.

### 3.5 OPTIMIZATION ANALYSIS

Assume that the generator input and output are of the same dimension, mapping $q(z) \sim \mathcal{N}(0, 1)$ to real data distribution $p_r$, look at it another way, it essentially learns $-(x_t - x_0)$ if we define $x_t \sim q(z)$ and $x_0 \sim p_r$, then sampling from $x_t$ with $-(x_t - x_0)$. Define

$$-(x_t - x_0) = x_0 - x_t = f_t(x_t)t \tag{21}$$

here $f_t$ is fitted by generator, so

$$x_0 - x_t = f_t(\theta, x_t)t \tag{22}$$

If we interpolate between $x_0$ and $x_t$, making $\Delta t \to 0$, have

$$x_{t+\Delta t} - x_t = f_t(\theta, x_t)\Delta t + g_t\sqrt{\Delta t}z, z \sim \mathcal{N}(0, 1) \tag{23}$$

where $g_t$ is called diffusion term. Eq. 22 and 23 tells us that diffusion model just divides $x_0 - x_t$ into several parts to learn compared to GANs. We multiply the LHS of Eq. 22 by a factor $\frac{1}{\sigma}$, here we can define $\frac{1}{\sigma} = \|x_t - x_0\|$, making easy for generator to learn with this normalization factor

$$-\frac{(x_t - x_0)}{\sigma} = f_t(\theta, x_t)t \tag{24}$$

the LHS of above equation is so-called score function according to VE-SED in Song et al. (2020), namely

$$-\frac{(x_t - x_0)}{\sigma} = s_\theta(x_t) \tag{25}$$

we add the correction score $C_\theta(x_t)$ after post-hoc discriminator reaches its optimal point. The adjusted model score can be formulated as

$$s_\theta(x_t) + \omega_c C_\theta(x_t) = \nabla \log p_f(x_t) + \omega_c \nabla \log \frac{p_r(x_t)}{p_f(x_t)} = \nabla \log \left[ (p_r(x_t)^{\omega_c})(p_f(x_t))^{1-\omega_c} \right] \tag{26}$$

by setting $\omega_c = 1$, we can reach the gradient of logarithm of real data probability density function, so we fix $\omega_c = 1$ if not stated otherwise.

## 4 MAIN EXPERIMENTS

In this section, we present the main experiments with regard to the combinations of proposed post-hoc discriminator guidance (PDG) and various mainstream data-efficient GANs on content-varying datasets ranging from $256 \times 256$ to $1024 \times 1024$ resolution. We choose FID (Heusel et al. (2017)) as our metric, for it is able to objectively measure the Kullback-Leibler divergence between real and generated samples. Besides, a new metric FID Gain=FID(After PDG) − FID(Before PDG) is come up with for better showing the performance of our method (negative values are normal in this case). The details concerning descriptions of baseline methods and datasets can be found in appendix.

### 4.1 MAIN ABLATION EXPERIMENTS

According to statistic data from Table 1, 2 and 3 carried on fifteen datasets, it is concluded that PDG is indeed able to close the KL distance between the true and generated distributions, simply by training an additional discriminator post-hoc, sampling via annealing Langevin dynamics. In fact, a post-hoc discriminator converges in less than 20 minutes on a Tesla V00 GPU, which is also due to the fact that the training of the discriminator is a classification task, rather than a minimax optimization. These findings highlight the potential of post-hoc discriminator as a supplementary method to cope with poor optimization at large time in adversarial training.

### 4.2 ABLATION EXPERIMENTS ON PD NUMBER OF EPOCHS

Fig. 3 illustrates how the FID Gain of post-hoc discriminator (PD) on different methods varies with the number of epochs for its discriminator training. We notice there are quite a few cases where the post-hoc discriminator falls into overfitting as the number of iterations increases, and this is independent of the choice of the baseline model. We suspect that this is caused by data quality imbalance in the real data and the generated data under data-limited condition, after all, the quality of the generated images also determines the speed of convergence.

### 4.3 ABLATION EXPERIMENTS ON PD TIMESTEP $T$

Here timestep $T$ denotes the number of iterations an image goes through, just like the number of samples $T$ for a diffusion model. From Table 4, we can see that the timestep $T$ is generally proportional to the FID Gain, but some cases show that the long time step leads to the degradation

Table 1: Ablation on various mainstream data-efficient GANs and FID (↓)(Heusel et al. (2017)) comparison on five datasets, bold font denotes the best result.

| Method | AF-Wild (146 imgs) | AF-Cat (160 imgs) | AF-Dog (389 imgs) | Obama (100 imgs) | Panda (100 imgs) |
|---|---|---|---|---|---|
| FastGAN | 16.5749 | 34.3854 | 53.4005 | 37.2040 | 9.4016 |
| +Ours | **16.4632 (-0.1117)** | **34.3784 (-0.007)** | **53.2726 (-0.1279)** | **37.2003 (-0.0036)** | **9.3823 (-0.193)** |
| FreGAN | 15.1452 | 34.0711 | 55.6129 | 36.9809 | 9.2764 |
| +Ours | **15.1212 (-0.024)** | **34.0673 (-0.0037)** | **55.5569 (-0.0560)** | **36.9777 (-0.0032)** | **9.1512 (-0.1252)** |
| Lecam | 21.5653 | 36.6119 | 56.8955 | 40.5946 | 9.8552 |
| +Ours | **21.3657 (-0.1996)** | **36.4129 (-0.1990)** | **56.8852 (-0.0103)** | **40.1138 (-0.1508)** | **9.4270 (-0.4282)** |
| MoCA | 17.5341 | 38.9585 | 57.4012 | 41.6329 | 11.2332 |
| +Ours | **17.4588 (-0.0753)** | **38.8919 (-0.0666)** | **57.3802 (-0.021)** | **41.2990 (-0.3339)** | **11.1806 (-0.0526)** |
| ProtoGAN | 17.8684 | 35.8733 | 54.4858 | 38.3622 | 9.8410 |
| +Ours | **17.7305 (-0.1379)** | **35.8570 (-0.0163)** | **54.2876 (-0.1982)** | **38.2027 (-0.1595)** | **9.6732 (-0.1677)** |

Table 2: Ablation on various mainstream data-efficient GANs and FID (↓)(Heusel et al. (2017)) comparison on five datasets, bold font denotes the best result.

| Method | Grumpy-Cat (100 imgs) | Fauvism (124 imgs) | Moongate (135 imgs) | Shells (64 imgs) | Skulls (96 imgs) |
|---|---|---|---|---|---|
| FastGAN | 25.6070 | 182.8983 | 117.4819 | 172.3005 | 103.0316 |
| +Ours | **25.2578 (-0.3492)** | **182.3601 (-0.5382)** | **117.4161 (-0.0658)** | **162.1135 (-0.7187)** | **100.7855 (-2.2461)** |
| FreGAN | 25.9469 | 175.8942 | 117.6969 | 133.4191 | 97.0861 |
| +Ours | **25.7263 (-0.2206)** | **173.5741 (-2.3201)** | **112.3349 (-5.3620)** | **131.2199 (-2.1992)** | **94.5485 (-2.5376)** |
| Lecam | 25.7847 | 66.7609 | 133.8787 | 178.0432 | 114.6667 |
| +Ours | **25.3522 (-0.4325)** | **66.6165 (-0.1444)** | **133.2262 (-0.6525)** | **170.7767 (-7.2665)** | **113.8363 (-0.8304)** |
| MoCA | 28.1053 | 168.0824 | 119.7157 | 153.9461 | 123.5183 |
| +Ours | **27.3458 (-0.7595)** | **167.9367 (-0.1457)** | **119.5381 (-0.1776)** | **145.9907 (-7.9554)** | **122.5855 (-0.9328)** |
| ProtoGAN | 25.7601 | 172.9304 | 121.9207 | 135.1166 | 97.5333 |
| +Ours | **25.6173 (-0.1428)** | **172.7905 (-0.1399)** | **121.7978 (-0.1229)** | **132.6274 (-2.4892)** | **97.4565 (-0.0768)** |

Table 3: Ablation on various mainstream data-efficient GANs and FID (↓)(Heusel et al. (2017)) comparison on five datasets, bold font denotes the best result.

| Method | CUB (100 imgs) | Flowers (100 imgs) | Nature landscape (100 imgs) | Place365-Standard (100 imgs) | ImageNet (100 imgs) |
|---|---|---|---|---|---|
| FastGAN | 140.4356 | 107.0951 | 48.6951 | 105.2352 | 252.4403 |
| +Ours | **139.1887 (-1.2469)** | **102.8725 (-4.2226)** | **48.6116 (-0.0835)** | **105.1985 (-0.0367)** | **252.4339 (-0.0064)** |
| FreGAN | 135.6631 | 67.6844 | 49.2108 | 87.1500 | 236.2905 |
| +Ours | **135.3548 (-0.3083)** | **65.4276 (-2.2568)** | **48.0652 (-1.1456)** | **87.1277 (-0.0222)** | **236.2134 (-0.0771)** |
| Lecam | 142.4079 | 124.4659 | 63.9420 | 107.9235 | 275.6394 |
| +Ours | **140.2693 (-2.1386)** | **117.6672 (-6.7987)** | **62.0156 (-1.9264)** | **106.9829 (-0.9406)** | **275.6143 (-0.0251)** |
| MoCA | 144.2144 | 87.7164 | 52.4950 | 98.0106 | 259.9405 |
| +Ours | **143.2271 (-0.9873)** | **86.0864 (-1.6300)** | **48.7353 (-3.7597)** | **96.3927 (-1.6179)** | **259.0767 (-0.8638)** |
| ProtoGAN | 143.8905 | 81.2860 | 47.0146 | 93.2369 | 261.6027 |
| +Ours | **142.8810 (-1.0095)** | **81.1078 (-0.1782)** | **47.0070 (-0.0076)** | **92.9925 (-0.2444)** | **261.5719 (-0.0308)** |

Table 4: Ablation on timestep $T$ and FID Gain(↓)(Heusel et al. (2017)) comparison on three datasets, bold font denotes the best result.

| Method | Grumpy-Cat (255 × 256) | | | | | Shells (512 × 512) | | | | | Flowers (1024 × 1024) | | | | |
|---|---|---|---|---|---|---|---|---|---|---|---|---|---|---|---|
| | T=5 | T=10 | T=15 | T=20 | T=25 | T=5 | T=10 | T=15 | T=20 | T=25 | T=5 | T=10 | T=15 | T=20 | T=25 |
| FastGAN | -0.3293 | -0.3469 | -0.1535 | **-0.3491** | -0.0834 | -6.8665 | -6.6561 | **-7.1870** | -6.6300 | -6.5589 | -4.1543 | **-4.2225** | -3.7452 | -3.8132 | -4.1090 |
| FreGAN | -0.0378 | -0.1199 | -0.1381 | -0.1886 | **-0.2205** | 0.7884 | -0.6015 | -1.1117 | **-2.1991** | -1.1493 | -1.8379 | -1.7894 | -1.7781 | **-2.2568** | -2.0348 |
| Lecam | -0.2766 | **-0.4325** | -0.3860 | -0.1817 | 0.0001 | -5.1391 | -5.9092 | **-7.2665** | -7.1326 | -6.9152 | -6.3237 | -6.5624 | -6.6256 | -6.7410 | **-6.7987** |
| MoCA | -0.2763 | -0.5149 | -0.6635 | **-0.7594** | -0.7550 | -7.5542 | -7.2929 | -7.3898 | **-7.7089** | -7.0208 | -1.2582 | -1.3024 | -0.7481 | **-1.6299** | -1.3045 |
| ProtoGAN | -0.0949 | **-0.1428** | -0.1280 | -0.1423 | -0.0580 | -1.2599 | -1.7558 | -2.4143 | -2.4288 | **-2.4892** | -0.0706 | -0.0705 | -0.1181 | -0.1578 | **-0.1782** |

of image quality, like the DDPM (Ho et al. (2020)) with a few thousand time steps is not as good as a thousand time steps.

## 5 PARALLEL WORKS

Concurrently, Kim et al. (2022) is the closest work to ours, it deceives a discriminator by adding an auxiliary term to the pre-trained score and helps diffusion models better score estimation in a complementary way. However, our method is specially designed for GANs' sampling improvement, offering a stable minimization optimization to reduce kullback-Leibler distance between generated images and real images, which avoids unstable adversarial training. Sadat et al. (2024) proposes independent condition guidance, by using the fact that conditional score function will be equivalent to the unconditional score when a conditional vector is independent of the input data. On the basis of CG, SDG (Liu et al. (2023)) extends the condition term to various semantic conditions, and guides the cosine similarity between images and texts in text-to-image generation. CFGHo &

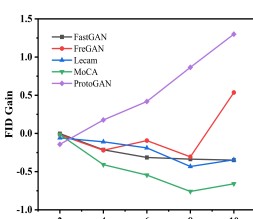 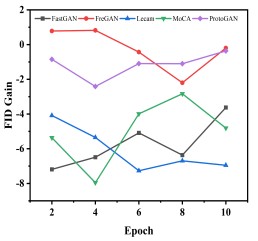 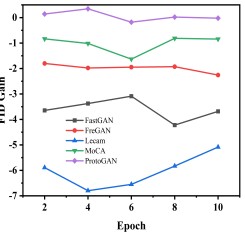

(a) FID Gain(↓) comparison on Grumpy Cat (Zhao et al. (2020)). (b) FID Gain(↓) comparison on Fauvism (Liu et al. (2020)). (c) FID Gain(↓) comparison on Shells (Karras et al. (2020a)).

Figure 3: We carry out post-hoc discriminator epoch ablation experiments, which accurately reflects the trend of FID Gain as the post-hoc discriminator's epoch number changes.

Salimans (2022) replaces the classifier gradient term used as a guide in CG by the difference between the conditional score and the unconditional score, thus eliminating the dependency on the classifier. Manifold constraint guidance (Chung et al. (2022)) finds that the projection-based correction operation makes the data deviate from the manifold, and proposes a correction term with manifold constraints. DSG (Yang et al. (2024)) regards the calculation of guidance as an optimization problem to minimize the guided-loss under the spherical Gaussian constraint, and calculates the corresponding closed-form solution.

In fact, all classifier guidance (CG) methods not just train a classifier, more importantly, the classifier must learn to classify noisy images, which is where CG is limited. Our model does not need to distinguish weather noisy images are realistic or not, greatly improving the convergence speed of the model.

# 6 DISCUSSION

## 6.1 CONCLUSIONS

We tear up the rulebook and study a post-hoc discriminator guidance for mitigating the negative effects induced by Nash non-equilibrium in GANs. We demonstrate that stochastic gradient descent (SGD) does not find the Nash equilibrium point in GANs' training because the Jacobian matrix of its associated gradient vector field has eigenvalues with zero real part and eigenvalues with large imaginary part. This method is more appealing than adversarial optimization as it is min-min problem instead of max-min GAN training. The optimal post-hoc discriminator predicts the score (density ratio gradient information) between the generated and true probability density function, to generate new samples that are more similar to the observed data. Our model is able to adapt to various off-the-shelf models, and extensive ablation experiments also verify that the Kullback-Leibler divergence between the true and generated samples can be reduced given an optimal discriminator.

## 6.2 LIMITATIONS AND IMPROVEMENTS

There are some training issues with regard to post-hoc discriminator, although our method is able to generalize to most datasets. Firstly, how well the post-hoc discriminator is trained determines the subsequent sampling performance, especially under limited-data condition, few data and long tail effects hinder the presence of optimal discriminator. The second issue concerns the generalization of post-hoc discriminator, which is influenced by many factors, including how well it fits the source data, how long it takes to train, how many samples it takes, and so on, for we are training the generated data with real images for classification, and then sampling with the generator. This requires the ability of the model to generalize. In the future, we will introduce classification networks for few-shot and long-tail effects, as well as self-supervised learning into post-hoc discriminator training for better performance.

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

# A APPENDIX

In this appendix, we mainly supplement the proofs of some lemmas in the main text section, as well as some additional explanations of the theory. In addition, the introduction of the datasets and the specific experimental details are also shown in this appendix, along with the pseudo-code for our model training and sampling. Finally, more data results are also presented for further understanding.

**Broader impact** The propose method achieves a new sampling aid in GANs, sidestepping the notorious adversarial training and building a homeomorphism mapping $f_\theta : \mathbb{R}_n \to \mathbb{R}_n$ between manifolds that expands topology basis in a topology space $\Lambda \in \mathcal{X} \cap \mathcal{Y}$ which consists of two open sets $\mathcal{X}$ and $\mathcal{Y}$ representing real data and generated data respectively. Besides, being capable of generating plausible and photorealistic images, our method brings potential issue of image abuse and fraud with the generated fake images. However, we believe that the rational use of such advanced technology can bring benefits to more fields like films and art production.

## A.1 ASSOCIATED GRADIENT VECTOR FIELD

Essentially, the optimization process of GANs is a process of minimizing some metric distance between two distributions, namely finding the suitable parameter $\theta$ to minimize $D(p_{data}, q_\theta)$ with adversarial method, where $D(\cdot, \cdot)$ denotes the wasserstein distance between real distribution and the distribution of data generated by generator with $\theta$.

**Definition 1** (Nash equilibrium) *Given two game players and their corresponding utility functions $g(\varphi, \theta)$ and $d(\varphi, \theta)$, $\varphi$ and $\theta$ represent discriminator and generator parameter respectively. For a deterministic parameter policy $(\overline{\varphi}, \overline{\theta})$, $\forall\, \theta$, $d(\varphi, \overline{\theta}) \leq d(\overline{\varphi}, \overline{\theta})$, $g(\overline{\theta}, \varphi) \leq g(\overline{\theta}, \overline{\varphi})$.*

The gradient field of any two differentiable two-player games can be defined as

$$\upsilon(\theta, \varphi) = \begin{bmatrix} \nabla_\varphi d(\theta, \varphi) \\ \nabla_\theta g(\theta, \varphi) \end{bmatrix} \tag{27}$$

$\upsilon(\theta, \varphi)$ is so-called associated gradient vector field defined by $d$ and $g$. Specially, GANs are typically zero-sum games, namely $d = -g$, let us take the derivative of $\upsilon$, have

$$\upsilon^{'}(\theta, \varphi) = \begin{pmatrix} \nabla_\varphi^2 d(\theta, \varphi) & \nabla_{\varphi, \theta} g(\theta, \varphi) \\ -\nabla_{\varphi, \theta} d(\theta, \varphi) & -\nabla_\theta^2 d(\theta, \varphi) \end{pmatrix} \tag{28}$$

based on the above definition, we have the following corollaries

**Corollary 1** *In zero-sum games, $\upsilon^{'}$ is negative semi-definite if and only if $\nabla_\varphi^2 d(\theta, \varphi)$ is negative semi-definite, as well as $\nabla_\theta^2 d(\theta, \varphi)$ positive semi-definite.*

**proof** $\forall\, w = (w_1, w_2)^T \neq 0$, $\exists$

$$w^T \upsilon^{'}(x) w = w_1^T \nabla_\varphi^2 d(\theta, \varphi) w_1 - w_2^T \nabla_\theta^2 d(\theta, \varphi) w_2 \tag{29}$$

hence, $w^T \upsilon^{'}(x) w < 0$ if and only if $w_1^T \nabla_\varphi^2 d(\theta, \varphi) w_1 < 0$ and $w_2^T \nabla_\theta^2 d(\theta, \varphi) w_2 > 0$ due to the arbitrariness of $w$.

The negative semi-definite case follows in the same way.

**proof end**

**Corollary 2** *In zero-sum games, $\upsilon^{'}(x)$ is negative semi-definite for any Nash equilibrium point $\overline{x}$. Instead, $\overline{x}$ is local Nash equilibrium point if $\overline{x}$ is stability point of $\upsilon(x)$ and $\upsilon^{'}(\overline{x})$ is negative semi-definite.*

**proof** If $\overline{x}$ is local Nash equilibrium point, $\nabla_\varphi^2 d(\overline{\theta}, \overline{\varphi})$ is negative semi-definite, $\nabla_\theta^2 d(\overline{\theta}, \overline{\varphi})$ is positive semi-definite. According to **Corollary 1**, for any Nash equilibrium point $\overline{x}$, $\upsilon^{'}(\overline{x})$ is always negative

semi-definite. Instead, if $\upsilon'(x)$ is negative definite, $\nabla^2_\varphi d(\overline{\theta}, \overline{\varphi})$ is negative semi-definite, $\nabla^2_\theta d(\overline{\theta}, \overline{\varphi})$ is positive semi-definite, so $\overline{x}$ is local Nash equilibrium point.

**proof end**

## A.2 RELATION WITH CLASSIFIER GUIDANCE

Classifier guidance aims to control the trained unconditional diffusion model through a classifier to achieve conditional generation, which can avoid re-training a conditional diffusion model and save computing resources. The forward process of diffusion model can be formulated as a stochastic differential equation

$$dx = f_t(x)dt + g_t d\omega \tag{30}$$

its corresponding reverse equation (Anderson (1982)) denotes

$$dx = \left(f_t(x) - \frac{1}{2}(g_t^2 + \sigma_t^2)\nabla_x \log p_t(x)\right) dt + \sigma_t d\omega \tag{31}$$

the key of condition generation lies in replacing $\log p_t(x)$ with $\log p_t(x|y)$. According to Bayes' theorem

$$\log p_t(x|y) = \nabla_x \log p_t(x) + \nabla_x \log p_t(y|x) \tag{32}$$

where $\nabla_x \log p_t(x) = -\frac{\epsilon_\theta(x_t,t)}{\overline{\beta}_t}$, thus

$$\nabla_x \log p_t(x|y) = -\frac{\epsilon_\theta(x_t,t)}{\overline{\beta}_t} + \nabla_x \log p_t(y|x) \tag{33}$$

combining Eq. 33 and 26, have

$$-\frac{\epsilon_\theta(x_t,t)}{\overline{\beta}_t} + \nabla_x \log p_t(y|x) \approx s_\theta(x_t) + \omega C_\theta(x_t) \tag{34}$$

In fact, we are training a classifier to improve the correlation of the generated results with the input signal $y$ if we set label of real data is 1, label of generated data 0. In other words, during the generation process, the post-hoc discriminator will pick out the samples with high classification confidence.

## A.3 TOPOLOGY PERSPECTIVE ON POST-HOC DISCRIMINATOR GUIDANCE

Here we want to explain our proposed method with topology, especially on the continuous mapping and homeomorphism happened in the change between before discriminator guidance (DG) and after discriminator guidance (DG). Constructing subspace is an important method to obtain new topological spaces from known ones. Define $(X, \mathcal{T})$ a topology space, $Y$ is the subset of $X$, let

$$\mathcal{T}|Y = \{U \cap Y | U \in \mathcal{T}\} \tag{35}$$

obviously $\mathcal{T}|Y$ is a topology of $Y$ (after DG), so-called limit of $\mathcal{T}$ on $Y$, meanwhile, $(Y, \mathcal{T}|Y)$ denotes the subspace of $(X, \mathcal{T})$ (before DG). Considering the boundedness of tensors, $(Y, \mathcal{T}|Y)$ also denotes closed subspace of $(X, \mathcal{T})$. Thus, we throw a proposition

**Proposition 3** *Assume $(Y, \mathcal{T}|Y)$ is the subspace of $(X, \mathcal{T})$, the subset $A$ of $Y$ is the closed set of $(Y, \mathcal{T}|Y)$ if and only if there exists closed set $F$ of $(X, \mathcal{T})$, making $A = Y \cap F$.*

**proof** If $A$ is the closed set of $(Y, \mathcal{T}|Y)$, $Y \backslash A$ is the open set of $(Y, \mathcal{T}|Y)$, so there exists the open set $U$ of $(X, \mathcal{T})$, making $Y \backslash A = Y \cap U$. Let $F = X n U$, $F$ becomes the closed set of $(X, \mathcal{T})$ and

$A = Y \cap F$. At this point, we prove the necessity of the proposition, and the proof of sufficiency is similar.

**Proposition 4** *Assume $X$ as a topology space, $Y \subseteq X$, satisfying: 1) If $\mathcal{B}$ is the base of $X$, the base of subspace $Y$ is $\{U \cap Y | U \in \mathcal{B}\}$; 2) If $\mathcal{A}$ is $y \in Y$ on one neighbourhood base of $X$, $\{V \cap Y | V \in \mathcal{A}\}$ denotes a neighbourhood base of $y$ on subspace $Y$.*

where $V$ is the open set of $X$, $V \subseteq U$. We know $Y$ is the subset of $X$, $A \subseteq Y$, define $cl_X A$, $int_X A$ as $A$'s closure and inner on $X$, $cl_Y A$, $int_Y A$ as $A$'s closure and inner on $Y$.

**Proposition 5** *Assume $Y$ is the subspace of $X$, $A \subseteq Y$, have 1) $cl_Y A = Y \cap cl_X A$; 2) $int_Y A = Y \supseteq int_X A$.*

**proof** Because $cl_X A$ is the closed set of $X$, $Y \cap cl_X A$ is the closed set of $Y$, also for $A \subseteq Y \cap cl_X A$, and $cl_Y A$ is the smallest closed set, so $cl_Y A \subseteq Y \cap cl_X A$. Furthermore, as $cl_Y A$ is closed set from $Y$, there exists $X$'s close set $F$ makes $cl_Y A = Y \cap F$. Also because $cl_X A$ is the smallest closed set $X$ includes $A$.

At this point, we have two topology spaces, denoting spaces before and after DG respectively.

**Definition 1** *Assume $X$ and $Y$ are two topology spaces, a mapping $f : X \rightarrow Y$ continues if given open set $U$ of $Y$ and the preimage $f^{\leftarrow}(U)$ of $U$ on $f$ is the open set of $X$.*

we define the mapping from $X$ to $Y$ is a continuous mapping, namely an equivalent characterization of continuous functions from $\mathbb{R}$ to $\mathbb{R}$.

**Proposition 6** *All constant mappings between topological spaces $X,Y$ are continuous, have 1) Given topological space $X$, identity mapping $1_X : X \rightarrow X$ is continuous; 2) If $f : X \rightarrow Y, g : Y \rightarrow Z$ are continuous mappings, composite mapping $g \circ f : X \rightarrow Z$ continues.*

1) is obvious, we focus on the proof of 2). Given open set $U$ of $Z$, because $g$ is continuous, $g^{\leftarrow}(U)$ is $Y$'s open set, $f$ is continuous, $(g \circ f)^{\leftarrow}(U) = f^{\leftarrow}(g^{\leftarrow}(U))$ is $X$'s open set, then $g \circ f$ is continuous.

Combining the previous conclusions, given topological space $(X, \mathcal{T})$, $Y$ is the subset of $X$, we have that $Y$'s sub space topology is the coarsest topology on $Y$ making inclusion mapping $i : Y \hookrightarrow X$ continuous, as well as the fact that mapping $f : Z \rightarrow Y$ continues if and only if $i \circ f$ continuous given topology space $Z$.

It is worth mentioning that the topology of $Y$'s subspace lets inclusion mapping continuous, namely $i : (Y, \mathcal{T}|Y) \hookrightarrow (X, \mathcal{T})$ is continuous. At the same time, If $\mathcal{S}$ is a topology of $Y$ and $i : (Y, \mathcal{S}|Y) \hookrightarrow (X, \mathcal{T})$ is continuous, $\mathcal{T}|Y \subseteq \mathcal{S}$. $\forall$ open set $U$ of $X$, $\exists$ $Y$'s subspace topology makes inclusion mapping continuous, for $i^{\leftarrow}(U) = \{y \in Y | i(y) \in U\} = U \cap Y$ is open set of $Y$'s subspace topology. Further, assume $\mathcal{S}$ is topology on $Y$ and $i : (Y, \mathcal{S}|Y) \hookrightarrow (X, \mathcal{T})$ continuous, $\forall V \in \mathcal{T}|Y, \exists X$'s open set $U$, resulting in $V = U \cap Y$, according to definition of subspace topology. Because $V = U \cap Y = i^{\leftarrow}(U)$, $V \in \mathcal{S}$ deduces $\mathcal{T}|Y \subseteq S$.

Of course, whether mapping $f : X \rightarrow Y$ is continuous decides on the topology of $X, Y$. But in practice, considering the gradient descent for neural networks, we fit this mapping with a neural network, which deduces the continuity of $f : X \rightarrow Y$. Next we figure out two topology spaces' boundary. The image tensors range from $-1$ to $1$.

**Proposition 7** *Given two topology spaces $X, Y$, $f : X \rightarrow Y$ continues, define sequence of $X$ $\{x_n\}$, if $\{x_n\}$ converges to $x$, $\{f(x_n)\}$ converges to $f(x)$. In other words, the continuous map preserves the limit of the sequence.*

$\forall$ neighbourhood $V$ of $f(x)$, for $f$ continues, $f^{\leftarrow}(V)$ is the neighbourhood of $x$, assume there exists $N \in \mathbb{N}$, resulting that $\forall n \geq N$, $x_n \in f^{\leftarrow}(V)$, namely $f(x_0) \in V$, then $\{f(x_n)\}$ converges to $f(x)$. Also because $\forall tensor \in [-1, 1]$, $|x_n| \rightarrow 1$ and $|f(x_n)| \rightarrow 1$.

On the other hand, mapping $f : X \rightarrow Y$ is one to one mapping, meaning that $X$ can be continuously deformed into $Y$, and the process is reversible (for we always can find a mapping $g : Y \rightarrow X$), so

**Definition 2** *Given two topology spaces $X, Y$, $f : X \rightarrow Y$ continues, assume there exists continuous mapping $g : Y \rightarrow X$, making $g \circ f = 1_X : f \circ g = 1_Y$, continuous mapping $f : X \rightarrow Y$ is homeomorphic mapping.*

Table 5: Ablation on timestep $T$ and FID Gain($\downarrow$) comparison on three datasets, bold font denotes the best result.

| Method | Panda (Zhao et al. (2020)) | | | | | Fauvism (Liu et al. (2020)) | | | | | AF-Wild (Choi et al. (2020)) | | | | |
|---|---|---|---|---|---|---|---|---|---|---|---|---|---|---|---|
| | T=5 | T=10 | T=15 | T=20 | T=25 | T=5 | T=10 | T=15 | T=20 | T=25 | T=5 | T=10 | T=15 | T=20 | T=25 |
| FastGAN | -0.0117 | -0.00913 | -0.0175 | -0.0151 | **-0.0193** | **-0.5382** | -0.1688 | 0.2298 | -0.3757 | 0.44864 | **-0.1116** | 0.0202 | 0.3115 | 0.5330 | 0.6043 |
| FreGAN | -0.1041 | **-0.1251** | -0.1030 | -0.0561 | -0.0297 | **-2.3201** | -1.3230 | -1.7703 | -1.2072 | -1.2220 | -0.0035 | **-0.0240** | -0.0232 | -0.0194 | -0.0067 |
| Lecam | -0.2309 | -0.3214 | -0.3669 | **-0.4281** | 0.3565 | 0.1421 | -0.0355 | 0.3363 | 0.0129 | **-0.1444** | **-0.1996** | -0.1523 | -0.1491 | -0.0922 | -0.0918 |
| MoCA | -0.0021 | -0.0143 | -0.0188 | -0.0473 | **-0.0526** | 0.5111 | **-0.1457** | 0.6026 | 0.4354 | 0.3386 | -0.0128 | -0.0264 | -0.0368 | **-0.0753** | -0.0435 |
| ProtoGAN | -0.2914 | -0.3423 | **-0.3891** | -0.3870 | -0.3034 | 0.0777 | 0.0229 | -0.0223 | -0.1161 | **-0.1399** | **-0.1379** | -0.0370 | 0.0846 | 0.0831 | 0.3126 |

In sampling, we fix DG hyper parameter $\omega_c = 1$, leading trajectory of sampling towards more realistic region. For $T = 1$ (the second iteration), we denote it by the Cartesian product space $X \times Y$, given two topology spaces $(X, \mathcal{T})$, $(Y.\mathcal{S})$, the Cartesian product between $X$ and $Y$ can be formulated as

$$\{U \times V | U \in \mathcal{T}, V \in \mathcal{S}\} \tag{36}$$

this is so-called $\mathcal{T} \times \mathcal{S}$, a topology generated by base. $X \times Y$ endows the topological space resulting from the product of $\mathcal{T}$ with $\mathcal{S}$ to be called the product space of $(X, \mathcal{T}), (Y, \mathcal{S})$.

**Proposition 8** *If $X$, $Y$ are nonempty topological spaces, define $p_X : X \times Y \to X$ and $p_Y : X \times Y \to Y$ are projections, we have the following conclusions:*

1) *$p_X : X \times Y \to X$ and $p_Y : X \times Y \to Y$ are continuous open mapping.*

2) *Production topology is the coarsest topology on $X \times Y$ which makes projection $p_X$ and $p_Y$ both continuous.*

3) *Given topological space $Z$ and mapping $h : Z \to X \times Y$, $h$ is continuous regarding production topology on $X \times Y$ if and only if $p_X \circ h : Z \to X$ and $p_Y \circ h : Z \to Y$ are continuous.*

We first prove the first conclusion. Given $X$'s open set $U$, $p_X^{\leftarrow}(U) = U \times Y$, so $p_X$ is continuous, then, given a basis $U \times V$ on $X \times Y$, $p_X^{\leftarrow}(U \times V) = U$, so $p_X$ is open mapping. On one hand, we know that projection $p_X$ and $p_Y$ are both continuous due to product topology on $X \times Y$, on the other hand, assume $\mathcal{R}$ is a topology on $X \times Y$ that makes projection $p_X$ and $p_Y$ both continuous, $\forall$ open set $U \subseteq X$ and $V \subseteq Y$, have

$$U \times V = (U \times Y) \cap (X \times Y) = p_X^{\leftarrow}(U) \cap p_Y^{\leftarrow}(V) \in \mathcal{R} \tag{37}$$

$\mathcal{R}$ is finer than the product topology $\mathcal{T} \times \mathcal{S}$, the coarsest topology on $X \times Y$ which makes projection $p_X$ and $p_Y$ are both continuous.

Here we begin to prove the third conclusion. If $h : Z \to X \times Y$ is continuous, composite mapping $p_X \circ h$, $p_Y \circ h$ are continuous. Given $Y$'s open set $V$ and $X$'s open set $U$, $h^{\leftarrow}(U \times Y) = (p_X \circ h)^{\leftarrow}(U)$ and $h^{\leftarrow}(X \times Y) = (p_Y \circ h)^{\leftarrow}(V)$ are both $Z$'s open sets, thus

$$h^{\leftarrow}(U \times V) = h^{\leftarrow}(U \times Y) \cap h^{\leftarrow}(X \times V) \tag{38}$$

so $h$ is continuous. We throw a theorem

**Theorem 2**$(X \times Y, p_X, p_Y)$ *Given product topology space $X \times Y$, $\forall Z, f : Z \to X, g : Z \to Y$, $\exists$ the only one continuous mapping $h : Z \to X \times Y$, making $f = p_X \circ h, g = p_y \circ h$.*

If $Z$ is a topological space, $f : Z \to X, g : Z \to Y$ are continuous, define $h : Z \to X \times Y$, $h(z) = (f(z), g(z))$, have $f = p_X \circ h, g = p_Y \circ h$. According to the third conclusion of **Proposition 8**, $h$ is continuous, and its uniqueness is obvious.

**Theorem 2** tells us that there exists a deterministic mapping from $Z$ to topological space after many times' discriminator guidance.

A.4 More visual data and tables

Please see Table 5, 6, 7, 8 and Fig. 4, 5.

Table 6: Ablation on timestep $T$ and FID Gain($\downarrow$) comparison on three datasets, bold font denotes the best result.

| Method | AF-Cat (Choi et al. (2020)) | | | | | AF-Dog (Choi et al. (2020)) | | | | | Obama (Zhao et al. (2020)) | | | | |
|---|---|---|---|---|---|---|---|---|---|---|---|---|---|---|---|
| | T=5 | T=10 | T=15 | T=20 | T=25 | T=5 | T=10 | T=15 | T=20 | T=25 | T=5 | T=10 | T=15 | T=20 | T=25 |
| FastGAN | 0.0015 | **-0.0070** | -0.0043 | -0.0037 | -0.0040 | -0.0910 | 0.0556 | **-0.1278** | 0.3063 | 0.3825 | **-0.0036** | 0.0054 | 0.0145 | 0.1208 | 0.0427 |
| FreGAN | 0.0010 | 0.0035 | 0.0014 | **-0.0037** | -0.0030 | -0.0077 | -0.0091 | -0.0083 | 0.0282 | **-0.0560** | -0.0003 | -0.0015 | -0.0003 | -0.0014 | **-0.0032** |
| Lecam | -0.1990 | -0.0497 | -0.0277 | 0.0829 | 0.1779 | 0.0038 | 0.0139 | **-0.0102** | 0.0023 | 0.0207 | -0.1620 | **-0.2260** | -0.1546 | -0.1094 | -0.0895 |
| MoCA | **-0.0001** | 0.0019 | 0.0043 | 0.0120 | 0.0094 | **-0.0209** | 0.0352 | 0.0149 | 0.0610 | 0.0699 | -0.1865 | **-0.3339** | -0.0341 | 0.0288 | 0.5148 |
| ProtoGAN | **-0.1632** | 0.0244 | 0.0018 | -0.0011 | 0.0974 | -0.0507 | -0.1034 | -0.1923 | -0.1841 | **-0.1982** | -0.0443 | -0.0425 | -0.0751 | -0.0887 | **-0.1595** |

Table 7: Ablation on timestep $T$ and FID Gain($\downarrow$) comparison on three datasets, bold font denotes the best result.

| Method | Moongate (Liu et al. (2020)) | | | | | Skull (Liu et al. (2020)) | | | | | Nature landscape (Liu et al. (2020)) | | | | |
|---|---|---|---|---|---|---|---|---|---|---|---|---|---|---|---|
| | T=5 | T=10 | T=15 | T=20 | T=25 | T=5 | T=10 | T=15 | T=20 | T=25 | T=5 | T=10 | T=15 | T=20 | T=25 |
| FastGAN | 0.4380 | 0.5047 | 0.5436 | **-0.0465** | 0.7885 | -1.7317 | -1.8548 | **-2.2461** | -2.2102 | -1.9601 | 0.1549 | **-0.0834** | 0.0168 | 0.3555 | 0.1057 |
| FreGAN | -4.4526 | -4.9770 | -5.007 | **-5.3620** | -4.2457 | -2.1380 | **-2.2761** | -2.5376 | -2.4153 | -2.2555 | -0.2996 | -0.7355 | -0.7275 | **-1.1456** | -1.0282 |
| Lecam | **-0.6524** | -0.0106 | 0.7474 | 1.5269 | 1.1231 | -0.5642 | -0.1160 | -0.2241 | **-0.8304** | -0.5292 | -1.6258 | -1.3905 | -1.4712 | -1.5547 | **-1.9264** |
| MoCA | 0.7241 | 0.1896 | 0.7780 | **-0.1775** | 0.5427 | **-0.9328** | -0.4503 | -0.7455 | -0.2889 | -0.8695 | **-3.7596** | -3.1451 | -3.6139 | -3.2358 | -3.6077 |
| ProtoGAN | -0.0294 | -0.0454 | -0.0675 | -0.0795 | **-0.1229** | 0.0067 | 0.0092 | 0.0080 | **-0.0005** | 0.0160 | **-0.0075** | 0.0034 | 0.0130 | -0.0022 | 0.0036 |

Table 8: Ablation on timestep $T$ and FID Gain($\downarrow$) comparison on three datasets, bold font denotes the best result.

| Method | Place365_Standard (López-Cifuentes et al. (2020)) | | | | | CUB (Wah et al. (2011)) | | | | | ImageNet (Deng et al. (2009)) | | | | |
|---|---|---|---|---|---|---|---|---|---|---|---|---|---|---|---|
| | T=5 | T=10 | T=15 | T=20 | T=25 | T=5 | T=10 | T=15 | T=20 | T=25 | T=5 | T=10 | T=15 | T=20 | T=25 |
| FastGAN | -0.0022 | 0.0017 | 0.0024 | 0.0045 | **-0.0046** | -1.1172 | **-1.2469** | -1.0001 | 0.8087 | 0.4585 | -0.0015 | 0.0318 | 0.0208 | 0.0375 | **-0.0063** |
| FreGAN | -0.0057 | **-0.0222** | -0.0046 | -0.0122 | -0.0214 | **-0.3082** | 0.1993 | 0.5201 | 0.3553 | 0.9626 | **-0.0770** | 0.0651 | 0.1483 | 0.1073 | 0.0446 |
| Lecam | -0.3593 | -0.5142 | -0.0684 | -0.7830 | **-0.9406** | -0.7459 | **-2.1386** | -1.3402 | -1.6490 | -1.2875 | 0.0041 | -0.0041 | 0.0125 | -0.0175 | **-0.0251** |
| MoCA | -0.7128 | -1.0183 | -1.3315 | -1.4557 | **-1.6179** | **-0.9873** | 0.0358 | 0.2030 | -0.0971 | 0.3310 | **-0.8638** | -0.8094 | 0.6974 | 0.1668 | 1.0332 |
| ProtoGAN | -0.0381 | -0.0400 | -0.0612 | -0.1549 | **-0.2444** | 0.4777 | 1.0943 | 0.1177 | **-1.0095** | 1.2232 | -0.0115 | 0.0042 | **-0.0307** | -0.0048 | 0.0557 |

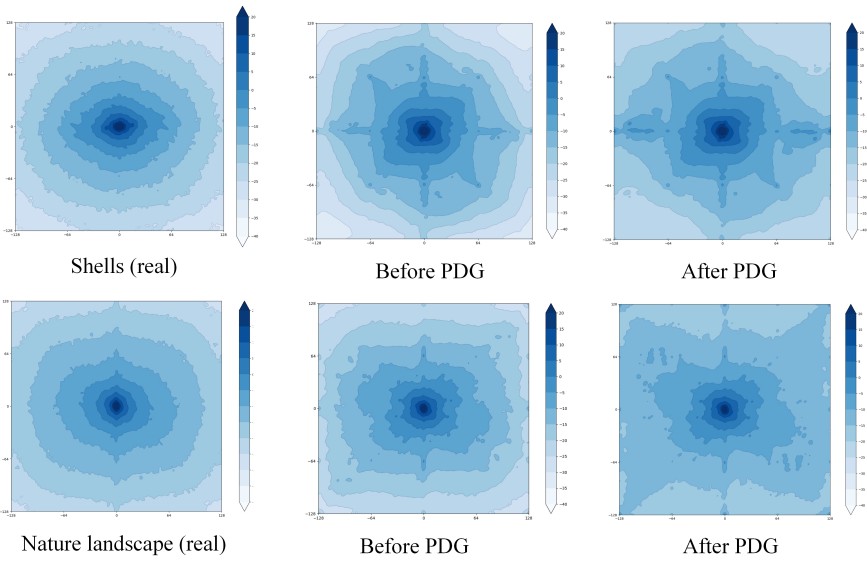

Figure 4: Comparison results of average 2D power spectrum on Nature landscape (Liu et al. (2020)) and Shells (Liu et al. (2020)). The average 2D power spectrum result for the real data is computed from all training data, and the results of our method and compared methods are computed from 2k generated images.

## A.5 MORE IMPLEMENT DETAILS

Our experiments are trained and evaluated on one Nvidia Tesla V100, we employ Adam as our optimizer with $\beta_1 = 0.5$ and $\beta_2 = 0.999$, meanwhile, $batchsize = 8$ and learning rates regarding generator and discriminator are $2e-4$. For all training sets, we calculate metrics by comparing $2,000$ generated images to the whole training set within $50,000$ iterations, and save checkpoint every $10,000$ iterations. **It is noted that we calculate the FID by comparing a batch of generated images and the same batch of generated images guided by the discriminator with the real image. This is a straightforward way to see if our approach works.** Furthermore, We choose

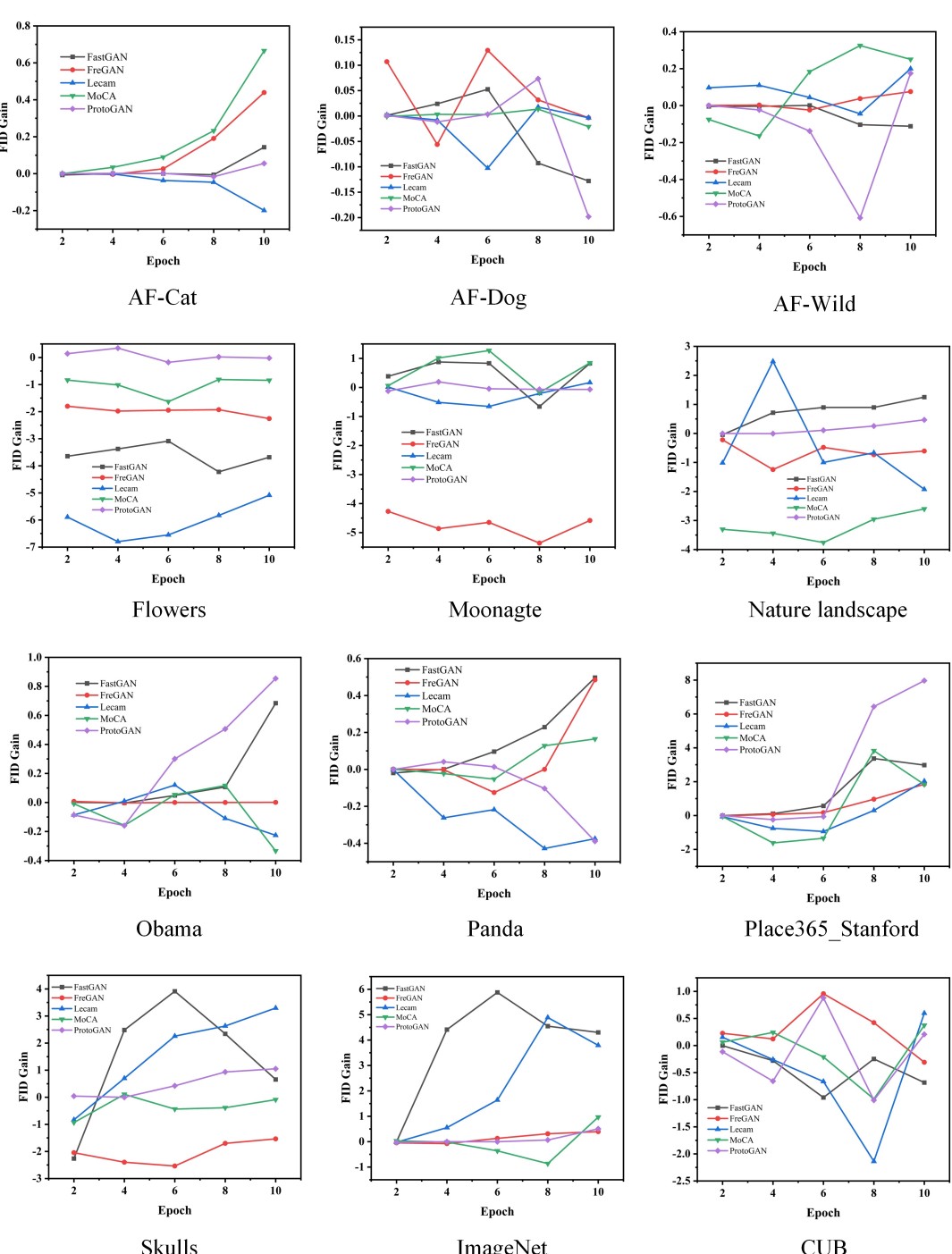

Figure 5: Post-hoc discriminator training epoch comparison results of FID Gain on the remaining 12 datasets.

the value of the original data with the lowest FID to compare with its post-discriminator-guidance value. Apart from training iteration, we keep hyper-parameters from other comparative networks unchanged. All experiments are run on PyTorch 1.7.1, and CUDA 10.2.

### A.5.1 IMPLEMENT DETAILS OF BASELINE METHODS

We reimplement all the baseline methods with their official code for fair comparisons. Notably, we select FastGAN (Liu et al. (2020)), FreGAN (Wang et al. (2022)), MoCA (Li et al. (2022)), Lecam (Tseng et al. (2021)) and ProtoGAN (Yang et al. (2023)) as our baseline models, these models all represent the main stream in data-efficient field of GANs. The ablation experiments demonstrate that our method is able to collocate different network structures. When combining our proposed techniques upon Lecam (Tseng et al. (2021)) and MoCA (Li et al. (2022)), we consistently keep all the details unchanged and add our proposed techniques to them. The coefficient of the regularization term is set as 0.1 following the original paper. Notably, we use the official code and recommended parameter of MoCA. All of our experiments are run on one Tesla V100 GPU, using PyTorch 1.7.1, and CUDA 10.2.

### A.5.2 IMPLEMENT DETAILS OF POST-HOC DISCRIMINATOR

The post-hoc discriminator network originates from Kim et al. (2022), which combines the encoder of Unet and a classification layer (e.g. fully connected layers and sigmoid activation function). We employ Adam as optimizer with fixed learning rate $3e - 4$ and weight decay $1e - 7$, $batchsize = 4$ throughout the all experiments, within 10 epochs (save checkpoints every two epochs) with $2,000$ generated images and $2,000$ real images (for datasets whose image number is lower than $2,000$, just copy themselves until image number satisfies), then utilize binary cross entropy loss function for gradient descent. In sampling, we set discriminator guidance $\omega_c = 1$, selecting the best results by different combinations of epoch network weights in training and number of iterations in sampling.

### A.6 POST-HOC DISCRIMINATOR SAMPLING, PYTORCH-LIKE

```
noise = torch.randn(args.batch, noise_dim).to(device)
g_imgs, _, _, _ = net_ig(noise, skips=None)
g_imgs_begin = g_imgs[0]

with torch.enable_grad():
 g_imgs_begin_to_grad = g_imgs_begin
 for inner_iter in range(inner_iteration):
  x_ = g_imgs_begin_to_grad.float().clone().detach().requires_grad_()
  logits = classifier(x_, t, sigmoid=True).view(-1)
  prediction = torch.clip(logits, 1e-5, 1. - 1e-5)
  log_ratio = torch.log(prediction / (1. - prediction))
  discriminator_guidance_score = /
  torch.autograd.grad(outputs=log_ratio.sum(), inputs=x_, retain_graph=False)[0]
  g_imgs = x_ + lambda_DG * discriminator_guidance_score
  g_imgs_begin_to_grad = g_imgs

for j, g_img in enumerate(g_imgs):
 vutils.save_image(g_img.add(1).mul(0.5),
 os.path.join(dist,
 '%d.png' % (i * args.batch + j)))  # , normalize=True, range=(-1,1))

for j, g_img in enumerate(g_imgs_begin):
 vutils.save_image(g_img.add(1).mul(0.5),
 os.path.join(dist_ori,
 '%d.png' % (i * args.batch + j)))
```

Listing 1: post-hoc discriminator sampling code

### A.6.1 IMPLEMENT DETAILS OF EVALUATION METRICS

Fréchet Inception Distance (FID) is a metric used to evaluate the discrepancy between the generative model and the real data distribution. It was proposed by Heusel et al. (2017) and is one of the widely used evaluation metrics nowadays.

The FID measures the difference between the generative model and the real data distribution by computing the Frechet distance between the two distributions. Frechet distance is a method to measure the distance between two distributions, which takes into account the mean and covariance matrix of the two distributions and can better describe the difference between the two distributions.

To compute the FID, a set of samples are first drawn from the real data distribution and the generative model, respectively, and then a pre-trained inception network is used to extract feature vectors from these samples. Next, the mean and covariance matrices of the two distributions are calculated and the Frechet distance between them is calculated to obtain the FID value. A smaller FID value means that the images generated by the generative model are closer to the true data distribution.

As an evaluation metric, FID is widely used in the training and evaluation of generative models. It can help us evaluate the quality of the generative model more accurately and select a better generative model. At the same time, FID is also an objective evaluation index, which can avoid the influence of human subjective factors on the evaluation results. The computation can be formulated as

$$FID = \|\mu_1 - \mu_2\|^2 + Tr(\Sigma_1 + \Sigma_2 - 2\sqrt{\Sigma_1 \Sigma_2}) \tag{39}$$

where $\mu_1$ and $\mu_2$ represents the mean vector of real data distribution and generated data distribution respectively, $\Sigma_1$ and $\Sigma_2$ denote the covariance matrix of the real and generated data, respectively. $Tr(\cdot)$ is trace of a matrix.

The FID value can be obtained by computing the mean covariance matrix of the two distributions and computing the Frechet distance between them. The smaller the FID value, the closer the generated image is to the real data distribution.

## A.7 DATASETS DESCRIPTION

AFHQ (Choi et al. (2020)) datasets contain 5k training images of animal faces with 512 resolution, respectively AF-Dog (389 imgs sampled from original data), AF-Cat (160 imgs sampled from original data) and AF-Wild (146 imgs sampled from original data). The dataset is made available under the Creative Commons BY-NC 4.0 license. In experiments, we uniformly compress them to $256 \times 256$ resolution.

100-shot (Zhao et al. (2020)) datasets contain various contents of images, and all the datasets contain 100 $256 \times 256$ training images (e.g. 100-shot-obama, 100-shot-panda and 100-shot-grumpy-cat). They are ideal for verifying the quality of the generation in low-shot scenarios.

BrecaHAD (Aksac et al. (2019)) dataset contains 162 images for breast cancer histopathological annotation and diagnosis. Its texture and content are complex, thus is suited for evaluating GANs'performance under limited data, facilitating the exploration of data-efficient GANs for downstream tasks of the medical field.

Moongate ($512 \times 512$ resolution 135 imgs), fauvism ($512 \times 512$ resolution 124 imgs), shells ($1024 \times 1024$ resolution 64 imgs), skulls ($1024 \times 1024$ resolution 96 imgs) and nature landscape (100 $1024 \times 1024$-resolution imgs sampled from original data) datasets come from Liu et al. (2020). These datasets include less than 150 images with different resolutions. Thus we adopt them for evaluating our model under limited data. We resize them to the closest resolution in implementation.

CelebA-HQ (Karras (2017)) is a high-quality face dataset developed by the Chinese University of Hong Kong and contains 30,000 images with 1024×1024 resolution. This dataset is a high-quality version of CelebA suitable for various computer vision tasks such as image generation, image super-resolution, image-to-image translation, etc. We randomly sample 100 images for training and metric computation.

Cub-200-2011 (Wah et al. (2011)) dataset is a fine-grained image classification dataset released by Caltech, which is an extended version of CUB-200 dataset. The dataset contains 11,788 images of 200 bird species, and each image is annotated with detailed information, including object bounding boxes, part locations, binary attributes, and subcategory labels. These images are often used for few-shot fine-grained image classification or detection tasks. Specifically, we randomly sample 100 images from it and resize them to $256 \times 256$ resolution for training and sampling.

Oxford 102 Flowers (Nilsback & Zisserman (2008)) Dataset is a flower collection dataset mainly used for image classification. It is divided into 102 categories totaling 102 flowers, where each category contains 40 to 258 images. The dataset was published in 2008 by the Department of Engineering Sciences, University of Oxford. Similarly, we randomly sample 100 images from it and resize them to $512 \times 512$ resolution for training and sampling.

The Places365 (López-Cifuentes et al. (2020)) dataset is a scene recognition dataset. It is composed of 10 million images comprising 434 scene classes. There are two versions of the dataset: Places365-Standard with 1.8 million train and 36000 validation images from K=365 scene classes, and Places365-Challenge-2016, in which the size of the training set is increased up to 6.2 million extra images, including 69 new scene classes (leading to a total of 8 million train images from 434 scene classes). we randomly sample 100 images from it and resize them to $256 \times 256$ resolution for training and sampling.

Flickr-Faces-HQ (FFHQ) (Karras et al. (2019)) is a high-quality image dataset of human faces, originally created as a benchmark for generative adversarial networks (GAN). The dataset consists of 70,000 high-quality PNG images at 1024×1024 resolution and contains considerable variation in terms of age, ethnicity and image background. It also has good coverage of accessories such as eyeglasses, sunglasses, hats, etc. The images were crawled from Flickr, thus inheriting all the biases of that website, and automatically aligned and cropped using dlib. Only images under permissive licenses were collected. Various automatic filters were used to prune the set, and finally Amazon Mechanical Turk was used to remove the occasional statues, paintings, or photos of photos. We randomly sample 100 images for training and metric computation.

The ImageNet (Deng et al. (2009)) dataset is a computer vision dataset created by Professor Fei-Fei Li at Stanford University. The dataset contains 14,197,122 images and 21,841 Synset indices. A Synset is a node in the WordNet hierarchy, which is again a set of synonym sets. The ImageNet dataset has been a benchmark for evaluating the performance of image classification algorithms. The ImageNet dataset is a large image dataset established to promote the development of computer image recognition technology. The 2016 ImageNet dataset already has over 10,000 images, each of which has been manually labeled. The images in the ImageNet dataset cover most of the types of images you will see in life. ImageNet was originally a dataset with over 1 million images. It contains a variety of images, and each image is associated with a label (category name). The ILSVRC image recognition competition using this huge dataset is held every year. Similarly, we randomly sample 100 images for training and metric computation.

