# OpenReview forum: "Post-hoc Discriminator Guidance For Data-efficient Image Generation Via Annealing Langevin Dynamics"
_ICLR.cc/2025/Conference — ICLR 2025 Conference Withdrawn Submission_

### Official Review · Reviewer_z9Vq · 2024-10-30

**Soundness:** 1
**Presentation:** 1
**Contribution:** 1
**Rating:** 1
**Confidence:** 5

**Summary:**

The authors propose an approach to carry out "post-hoc" discriminator guidance on the output of a GAN generator.

**Strengths:**

I do not find any strengths in this paper

**Weaknesses:**

There are several issues with this manuscript. Ranging from the writing, to the content, to the technical contribution. Overall, I suspect a major part of this has either been written by an LLM, or translated by one, as the language and writing are generally gibberish and inconsistent. Overall, I vote for a strong reject for this paper.

**Wording**: The language of the paper is terrible, and most things cannot be understood. To begin with, the idea of Nash “non-equilibrium” does not even make sense. Words such as “adversarial optimization,” the "GANs for minimax optimization, like the spin of a particle,” “unilateral optimization,” the “throw”ing of prepositions and theorems, using “$det(F)$, where $det(\cdot)$ denotes eigenvalue” all seem very weird at best, and wrong at worst. I could go on with examples from the paper, but I do not find it necessary to further make my point.

**Content**: The paper does not present a cohesive story. The sections appear to be random, and stapled together. The reason why the paper choses to prove the convergence of SGD, linking to contrastive mappings and what not, is beyond me.

**Novelty**: There is nothing novel about this work. Firstly, to cite Kim et al., who proposed discriminator guidance, only in Page 8 (L425) as a passing remark is disingenuous of the authors, especially when everything including the notation of $C_{\theta}$ for the discriminator guidance term is taken from this work. Besides this, there have been tons of works in the past year that have carried out discriminator guidance, e.g., [1,2,3,4] just to name a few, with a formulation equivalent to the presented Eqn. (14) present in both [2,3].

[1] Yi et al., “MonoFlow: Rethinking divergence GANs via the perspective of differential equations,” ICML 2023,
[2] Asokan et al., “GANs Settle Scores,” arXiv, 2023
[3] Franceschi et al., “Unifying GANs and Score-Based Diffusion as Generative Particle Models,” NeurIPS 2023
[4] Zhang et al., “DiffFlow: A Unified SDE for Score-Based Diffusion Models and Generative Adversarial Networks,” Openreview

**Methodology**: The analysis of casting the discriminator as an EM algorithm, only to introduce the hidden variable that mimics the role of the classification task of the discriminator seems roundabout. At the end of the day, the discriminator seems to be just any other GAN discriminator. The algorithm also does not make sense. How exactly do we get the generated dataset in Algorithm 1, to be able to train the “post-hoc” discriminator. If it is through a GAN-based process, how was that generator trained? Using a regular discriminator? If it is via a diffusion process, this paper is a blatant rip-off of Kim et al., with nothing new to show.

**Experimentation**: The experimental validation is unconvincing. The choices of datasets and baselines is not very clear, and sometimes erroneous. Maybe I am wrong, but I could not find the word “Shell” in Kerras et al., (2020), which this paper cites as the source for the “Shell” dataset. The obtained metrics are very poor, in the range of 200+, when SOTA FIDs are $\mathcal{O}(1)$, and the improvements are sometimes in the third decimal place!

The list of weaknesses above, in my opinion, is still incomplete. I do not wish to sound harsh, but I wish to make the point that I don’t think it is fair for reviewers to spend more time crafting a review for a paper than the time it appears that the authors have spent crafting the paper.

**Questions:**

See Weaknesses

---

### Official Review · Reviewer_UkJ1 · 2024-10-31

**Soundness:** 2
**Presentation:** 1
**Contribution:** 2
**Rating:** 3
**Confidence:** 4

**Summary:**

The authors propose a post-hoc discriminator guidance (PDG) method so that they can sample from the more realistic regions of GAN output. They also utilize annealing Langevin dynamics to sample to reduce the Kullback-Leibler divergence between the true and generated samples. Though the experiments show some improvement, the incremental numbers cannot quiet prove its effectiveness.

**Strengths:**

- It's interesting to learn or sample from the high fidelity areas in the GAN's output space based on the overlapping with the real distribution.

**Weaknesses:**

- The paper is hard to follow:
    - I recommend the authors to reformulate the paper to make it more readable. For example, you might firstly give the post-hoc training method, then dive deep into the detailed explanations.
    - For the mathematical symbols, define them in first usage. For example, $s_\theta(x_t)$ and $\omega_c$ are not defined in line 60. This also happens for $v'$ and $h$ in Proposition 1, and what is $F^{(k)}$ in Theorem 1?
    - What's the meaning of $y$ in equation (9)?
    - what is $\tilde{p}(x)$ in line 160, and is this a typo $p(x, y) = p(x, y)\tilde{p}(x)$?

- The experiments are not promising, less than $1\%$ improvement cannot prove the efficient of the method.

**Questions:**

Please see weakness.

---

### Official Review · Reviewer_3gi7 · 2024-11-01

**Soundness:** 3
**Presentation:** 2
**Contribution:** 2
**Rating:** 3
**Confidence:** 4

**Summary:**

The proposed method, post-hoc discriminator guidance (PDG) aims to take an alternate route for Nash non-equilibrium issue in GANs’ training. This method introduces an additional discriminator that gives explicit supervision with regard to gradient of density ratio between real and fake probability density function, steering the sample path towards more realistic regions in a post-hoc
way.
The discriminator was trained after adversarial optimization, making post-hoc
discriminator training stable and fast to converge. In generation process, annealing Langevin dynamics sampling with density ratio score reduces the KL divergence between the true and generated samples. Given an optimal discriminator, the method can improve the sampling quality of various off-the-shelf models on the web without retraining required. Extensive experiments validate the advancements and effectiveness of PDG on content-varying data-limited datasets.

**Strengths:**

The article introduces a post-hoc discriminator guidance method to take an alternate route for Nash non-equilibrium issue in GANs’ training. The discriminator was trained after adversarial optimization, making post-hoc discriminator training stable and fast to converge. In generation process, annealing Langevin dynamics sampling with density ratio score reduces the KL divergence between the true and generated samples. Given an optimal discriminator, the method can improve the sampling quality of various off-the-shelf models on the web without retraining required.

**Weaknesses:**

1. The empirical results seem too trivial to support the claim from the article that “the method can improve the sampling quality of various off-the-shelf models on the web without retraining required.”

2. The article lacks a theoretical analysis explaining why a gradient penalty is necessary to stabilize Langevin annealing and the convergence of Langevin-MCMC with density-ratio scores.

**Questions:**

1. In Tables 1-4, the FID GAIN in most datasets is minimal. I believe these results do not validate the effectiveness of your PDG.
2. In Figures 2, 3, and 5, the FID gain with different baselines is too small and complex. I believe the authors should explain why the PDG with different baselines shows such varying tendencies in FID gain. Some increase while others decrease.

3. I recommend that the author include a theoretical analysis of why the gradient penalty is necessary for stable Langevin annealing and discuss the convergence of Langevin-MCMC with the density-ratio score.

4. In Section A.3, there are numerous definitions and propositions without careful clarification. I advise the author to reorganize this content.

5. I recommend that the author visualize generated data for a single class label to effectively demonstrate the improvement in diversity.

6. I recommend that the author include a comparison with a similar post-hoc method using GANs in the experiments.

---

### Official Review · Reviewer_HmJc · 2024-11-04

**Soundness:** 3
**Presentation:** 2
**Contribution:** 2
**Rating:** 5
**Confidence:** 4

**Summary:**

To improve generative modeling in data-limited datasets using GANs, this paper proposes optimizing the GAN generator through post-hoc discriminator guidance. Specifically, the optimized discriminator provides a density ratio score to reduce the Kullback-Leibler divergence by iteratively running annealed Langevin dynamics sampling. Experimental results across various datasets demonstrate the effectiveness of the proposed post-hoc discriminator guidance.

**Strengths:**

Using post-hoc discriminator guidance to enhance modeling performance is indeed a reasonable approach. This method effectively addresses sampling bias between the target density and the modeled density through the gradual correction enabled by annealed Langevin dynamics sampling. By iteratively refining the sampling process, it aligns the generated samples more closely with the true data distribution.
This paper conducts numerous experiments on various datasets, all of which show some degree of improvement.

**Weaknesses:**

1. This paper has a strong connection with [1], as both employ additional discriminator guidance to improve sampling by rectifying sampling bias. The only significant difference is the generator utilized by the GAN in this paper. Therefore, the main contribution may be considered limited.

2. Is it really useful to use more sampling steps to improve GAN modeling? If the goal is solely better performance, why not directly employ diffusion models for sampling, given their proven modeling capability?

3. In my opinion, Section 2 shows limited connection with the methodology section. Furthermore, the theoretical analysis is somewhat simplistic and closely related to [1].

4. The experimental results are marginal improvement on various datasets.

5. It is suggested that the format of references should be uniform.

6. The writing should follow the policy because the pseudocode in the appendix is beyond the boundaries.

[1] Dongjun Kim, Yeongmin Kim, Se Jung Kwon, Wanmo Kang, and Il-Chul Moon. Refining generative process with discriminator guidance in score-based diffusion models. arXiv preprint arXiv:2211.17091, 2022.

**Questions:**

1. How does the time $t$ relate to the discriminator guidance when the training only inputs clean images?

2. Can the performance be improved if use larger batchsize?

3. Could you compare the sampling latency between the baseline and the implementation of the proposed framework?

---

### Note · Authors · 2025-01-22

I have read and agree with the venue's withdrawal policy on behalf of myself and my co-authors.